# Traumatic Hip Dislocation Associated with Proximal Femoral Physeal Fractures in Children: A Systematic Review

**DOI:** 10.3390/children9050612

**Published:** 2022-04-25

**Authors:** Oana Haram, Elena Odagiu, Catalin Florea, Iulia Tevanov, Madalina Carp, Alexandru Ulici

**Affiliations:** 1Department of Pediatric Orthopedic Surgery, “Grigore Alexandrescu” Clinical Emergency Hospital for Children, 011743 Bucharest, Romania; oanamharam@gmail.com (O.H.); ciob_elena91@mail.ru (E.O.); catalind.florea05@gmail.com (C.F.); iulia.tevanov@gmail.com (I.T.); alexandruulici@yahoo.com (A.U.); 211th Department, Carol Davila University of Medicine and Pharmacy, Bulevardul Eroii Sanitari nr. 8, Sector 5, 050474 Bucharest, Romania

**Keywords:** traumatic hip dislocation, transphyseal fracture, avascular necrosis, open reduction, child

## Abstract

Traumatic hip dislocation might lead to serious complications and a poor outcome. Fortunately, it is a rare condition in pediatric patients. The purpose of this study is to establish and describe the complications caused by hip dislocations associated with transphyseal femoral neck fractures. Therefore, we conducted a literature review that resulted in 11 articles, including 32 patients, older than 10 years of age, suffering from traumatic hip dislocation associated with a transphyseal femoral neck fracture. We presented a case series of three patients with hip fracture-dislocation treated in our clinic that were also evaluated and included in the study. For the 35 patients included in the study group, the percentage of avascular osteonecrosis after hip fracture-dislocation was 88.57%. Traumatic hip dislocation associated with transphyseal femoral neck fracture is a rare condition and has a poor prognosis because of the high incidence of femoral head avascular necrosis (AVN). Reduction should be attempted within six hours the from injury, but this may not minimize the risk of AVN if transphyseal separation occurs. The approach may influence the development of AVN; lateral approach of the hip with great trochanter osteotomy seems to have the lowest number of cases of AVN.

## 1. Introduction

Traumatic hip dislocation (THD) is a rare condition in children, occurring in 3% of all joint dislocations [1]. Considering the relation between the displaced femoral head and the acetabulum, there are four types of traumatic hip dislocation: anterior, posterior, central and inferior [2]. The most common type of dislocation is posterior hip dislocation, which occurs approximately nine times more often than the anterior type [3]. In the pediatric age group, hip dislocation can be caused by high-energy trauma, such as motor vehicle accidents, falls from height, as well as low-energy trauma and same-level falls [4]. Low-energy trauma, such as slips, can cause these injuries in children under five years of age because of the hyperlaxity of the periacetabular structures. In older children approaching skeletal maturity, hip dislocation resembles that of an adult because of the high-energy trauma involved [5,6]. The growth plate can be weakened during dislocation [7,8]. Fractures of the femoral neck are usually more common in older patients but can be found in younger patients with different systematic diseases as well [9].

Pediatric hip dislocation with transphyseal femoral neck fracture is a rare condition, accounting for less than 1% of all hip injuries [9]. Femoral head and neck fractures comprise less than 8% of all hip fractures and 1% of all pediatric fractures [10]. The Delbet and Colonna classification of femoral neck fractures is based on the location of the fracture. According to this, the least frequent type of fracture is type I (transphyseal separation) and it might lead to femoral head osteonecrosis [9,11].

The treatment of pediatric hip fracture-dislocations is determined by the fracture type, patient’s age and the time span between the traumatic event and hospital presentation. It consists of pain management and urgent reduction in the operating room (OR). Open reduction of the dislocation and the fracture and internal fixation of the physeal separation is needed, followed by cast immobilization [4]. For children under 10 years of age, it is recommended to apply a spica cast in abduction for 3–4 weeks after reduction. After the age of 10 and for adolescent patients, the recommended period of cast immobilization after reduction is between 6 and 12 weeks [5].

THD is a medical emergency that requires immediate orthopedic assessment and reduction [12]. Because of the high risks of AVN it is necessary to look attentively at the blood supply of the femoral head. The medial femoral circumflex artery, which emerges from the profunda femoris artery, provides the major blood supply to the femoral head [12]. During reduction, the physician must be cautious about the danger of separating the capital femoral epiphysis (CFE) from the femoral neck; this may result in affecting the biomechanics of the physis or CFE. The CFE might be separated from the femoral neck while it is reduced in the acetabulum [13]. Partial reduction is likely to cause acute vascular disruption that affects the lateral circumflex artery and triggers irreversible blood loss to the CFE [14].

Early reduction is important to minimize the risk of ischemic (avascular) necrosis in this type of lesion [15]. The possible causes of femoral head and neck osteonecrosis are vascular shearing, interruption of the blood supply and local ischemia [9]. In the treatment of THD, early diagnosis is very important and performing a closed reduction as soon as possible is mandatory. Open reduction is indicated for the following cases: impossibility of a closed reduction or concentric reduction, and fracture-dislocation [2].

THD associated with transphyseal femoral neck fracture is related to a high rate of complications [2]. These complications are avascular necrosis, premature epiphyseal closure, nonunion, soft tissue loss injuries in open fracture-dislocation that requires negative pressure wound therapy [16], leg-length discrepancy, coxa vara and associated angular deformity [9].

After having a poor outcome treating the three patients in our clinic, we wanted to better understand this type of injury in order to find the best treatment with a better outcome. Hence, we performed this systematic review, and we studied the treatment and prognostic of complications from the literature cases.

## 2. Materials and Methods

We conducted the research on PubMed, Google Scholar and Web of Science, using the following keywords: traumatic hip dislocation in children, fracture-dislocation of the hip in children and dislocation of the hip associated with transphyseal fracture in children. The total number of articles with these criteria was 617. We used Covidence to review all these articles. The inclusion criteria were children with ages between 10 and 18 years, hip dislocation with physeal separation and articles published after 1990. The exclusion criteria were age under 10 years or over 18, patients with THD only or patients with transphyseal femoral head and neck fracture only, associated systemic conditions, wrong study design and articles published before 1990. We also excluded all the articles in languages other than English. The total number of articles included in our study was 11 (Figure 1, Table 1).

Two reviewers, M.C. and O.H., worked independently to assess the articles. The data was collected from these articles and are mentioned in Table 1 and Table 2.

We retrospectively evaluated three patients with THD associated with transphyseal femoral neck fracture treated in our clinic. We collected our information from medical charts, computer data and Pacs database. All three patients were boys, aged between 12 and 14 years. For each case, the mechanism of injury involved high-energy trauma. They all presented with posterior hip dislocation associated with transphyseal femoral head fracture. The diagnosis was established after physical examination and imagistic investigations. The patients underwent open reduction and internal fixation with Kirschner wires. One patient associated posterior column fracture of the acetabulum (Figure 2 and Figure 3).

All the cases were evaluated clinically and radiologically first at 2 weeks, then at 1, 2, 3, 4, 5, 6, 12 months, and last follow-up. All the patients developed osteonecrosis within an average of 6 months.

The main goal of our study was to determine the rates of developing osteonecrosis after this type of injury. The secondary goal was finding the correlation between osteonecrosis and patient age, sex and associated acetabular fracture.

## 3. Results

We have evaluated 35 patients, 32 from the 11 articles and 3 patients from our clinic. The mean age at diagnosis was 13 years and 2 months (ranging from 10 to 16 years). There were 5 females and 30 males. The most frequent injury mechanisms were road accidents (16 patients) and sport accidents (16 patients). There were 29 posterior dislocations and 4 anterior dislocations; for 2 cases, the type of dislocation was not stated. The time span between diagnosis and reduction varied from 5 h to 7 days for 16 patients; for the other 19 patients, it was not documented.

Because of the high trauma mechanism of injury, eight patients had other associated lesions (acetabulum fractures, peroneal nerve paralysis, distal radius fracture, femoral head fracture, bilateral tibiae and fibular fracture, multiple trauma severe head injury). The recommended treatment for traumatic hip dislocation associated with transphyseal femoral head fracture is open reduction and internal fixation. In 34 cases, the treatment was open reduction and internal fixation. In one case, treatment consisted of closed reduction and cast immobilization because of multiple trauma and severe head injury.

The approach for open reduction is according to the type of dislocation. In our group, the most often used approach was the posterior one (14 cases). Other approaches were posterolateral (9 patients), anterior (4 patients), lateral approach with greater trochanteric osteotomy (4 patients) and anterolateral (2 patients).

The most frequent complication after traumatic hip dislocation associated with transphyseal femoral head fracture was avascular necrosis, seen in 31 patients. Four patients did not develop AVN; three of them were treated by open reduction through a lateral approach with greater trochanteric osteotomy. The percentage of osteonecrosis after traumatic hip dislocation associated with transphyseal femoral neck fracture was 88.57%. Other complications were head subluxation (two patients), physeal closure (one patient), chondrolysis (one patient) and septic arthritis (one patient from the thirty-one patients who suffered from AVN). The fastest onset of AVN was 2 months and the latest was 72 months.

## 4. Discussion

Summarizing the results’ findings, we observed that AVN is the most common complication, and it has a rate of 88.57%. From the four patients that did not develop AVN, three were treated using open reduction through a lateral approach with greater trochanteric osteotomy.

Osteonecrosis of the femoral head can be influenced by the following factors: hip dislocation associated with transphyseal fracture, type of fracture, type of displacement, patient’s age and the time between trauma and surgery [9,17].

According to Trueta, there are five phases of femoral head vascularization: phase one—at birth, phase two—infantile, phase three—intermediate, phase four—pre-adolescent, phase five—adolescent. In the first phase, there are two sources of vascularization: one horizontal, which emerges from the lateral part of the head, and a vertical one, which comes from the top of the shaft. Phase two takes place from four months to four years of age, and, during this time, the main vascularization source is represented by the metaphyseal blood vessels; another important source is the lateral epiphyseal vessels; there is no blood supply coming from the ligamentum teres. In the third phase (four to seven years of age), the physis behaves as a wall between metaphysis and epiphysis. The lateral epiphyseal vessels are the only supply of blood for the femoral head. In this phase, there are no vessels that penetrate the epiphysis. Phase four (after about eight to ten years) is characterized by the fact that the physis continues to act like a barrier, but there are vessels that emerge from the ligamentum teres, penetrate the epiphysis and form anastomosis. Phase five is the last phase in which the adult stage of vascularization of the femoral head begins to form; this vascularization is represented by the anastomosis between lateral epiphyseal vessels, metaphyseal vessels and ligamentum teres vessels. This anastomosis penetrates the growing plate [18]. It is possible that this is a reason why a transphyseal femoral head fracture in adolescents leads to osteonecrosis because of the shearing of the vessels and local ischemia.

Regarding the complications, proximal femoral physeal damage can lead to limb length discrepancy and angular deformities in the femoral neck in children under 12 years of age [19]. In children over the age of 12, the most common complication is represented by coxa magna. Physeal damage can manifest after trauma, fractures, reactive hyperemia and synovitis [20].

Post-traumatic hip dislocation represents a medical emergency and should be reduced within six hours under anesthesia to minimize the risk of AVN [6]. The risk of osteonecrosis is 20 times higher in pediatric patients with delayed intervention, when compared to closed reduction within six hours from the injury [21]. In the case of associated transphyseal femoral head fracture, the time span until reduction does not seem to influence the development of AVN; two of the cases with no AVN had more than 6 h before reduction and for the other two cases we have no data.

When post-traumatic hip dislocation occurs in adolescents with open growth plates, the presence of the transphyseal fracture of the femoral head should be suspected. The indication, according to Herrera-Soto et al. [7], Kenon et al. [15] and Hougaard et al. [22], is reduction under general anesthesia, muscle relaxation and fluoroscopic guidance for better visualization. The femoral head should be secured with a pin before reduction to prevent the proximal femoral epiphysis from moving.

When a transphyseal fracture occurs, the proximal femoral physis is also damaged. According to Odent et al. [8], the physeal plate excision is useful because the damaged physis behaves as an obstacle for the revascularization of the femoral head. Also adding an additional bone graft, after screw removal, offers mechanical support for revascularization and remodeling of the affected femoral head.

In our study, four patients did not associate AVN. Three of these patients were treated by an open reduction through a lateral approach with greater trochanteric osteotomy. According to Van Nortwick et al. [23] and Schoenecker et al. [24], the Ganz surgical technique is useful in preserving the vascularization of the proximal epiphysis of the femoral head because it protects the medial femoral artery circumflex; it also offers a better visualization for an easier anatomic reduction.

Open reduction is recommended for a nonconcentric reduction, displaced femoral head or neck, failed closed reduction, acetabular fractures or dislocations accompanying physeal injury [25,26]. During the reduction of the hip, the muscular resistance and reduction maneuvers may cause an iatrogenic detachment fracture, separating the femoral epiphysis from the neck. Thus, the possibility of converting the closed reduction to an open one and complete muscle relaxation is most helpful [20].

According to Palencia J. et al. [10], Forlin et al. [27] and Başaran et al. [2], there is a higher possibility of a child older than 10 years developing AVN after a traumatic hip dislocation compared to younger children. Judging by this, the risk of developing AVN in all 31 patients was increased due to older age and the associated transphyseal fracture.

We used partial threaded pins in all the cases in our clinic. Pins may influence the appearance of osteonecrosis, but in our study, there is not sufficient data to have a statistically significant conclusion; thus, there is a need for further research. In the case of SCFE, from the literature, there is no difference regarding AVN between cases treated with pins and cases treated with screws, but pins have a higher complication and revision rate.

The risk of AVN is not impacted by a lack of weight-bearing or a prolonged period of immobilization [2,19]. It is important to remember that in the cases of transphyseal fracture associating a dislocation, the incidence of AVN can reach 100% [2].

There are two major pathologies responsible for AVN in adolescents, affecting the femoral head vascularization. There is a high risk of AVN for severe slipped CFE, even without the hip dislocation [28].

The results of our study show that the percentage of femoral head osteonecrosis after traumatic hip dislocation associated with transphyseal femoral neck fracture is 88.57%. The osteonecrosis is not influenced by the following attributes: age, type of trauma and interval between injury and surgery.

We had four patients that were treated with an open reduction through a lateral approach with greater trochanteric osteotomy; one of them developed AVN. Ganz surgical technique is a better approach for this type of injury because the vascularization of the proximal femoral epiphysis is protected. This technique has a greater learning curve and requires experienced hip surgeons. It is recommended for the physician to inform the parents that after this type of injury, the prognostic is poor and could lead to total hip arthroplasty.

This review has some limitations; this study does not have a registered protocol and it does not include a conference abstract and other similar literature.

## 5. Conclusions

Traumatic hip dislocation associated with transphyseal femoral neck fracture in children is a rare injury with a very poor prognosis, and a high incidence of AVN. Reduction should be attempted within six hours from the injury, but this may not minimize the risk of AVN if the transepiphyseal separation occurs. A possible approach, with a lower risk of AVN, is the lateral one with greater trochanteric osteotomy because the proximal femoral epiphysis vascularization is preserved. For older children with hip dislocation, it is necessary to prophylactically fix the femoral head before attempting reduction.

## Figures and Tables

**Figure 1 children-09-00612-f001:**
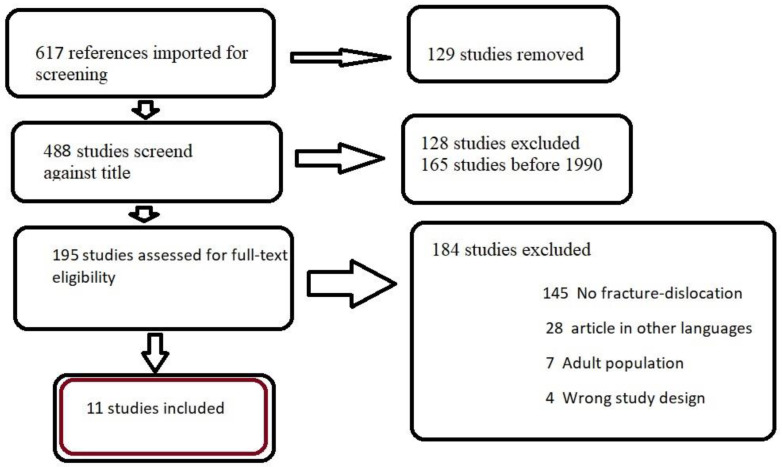
Literature review flow chart.

**Figure 2 children-09-00612-f002:**
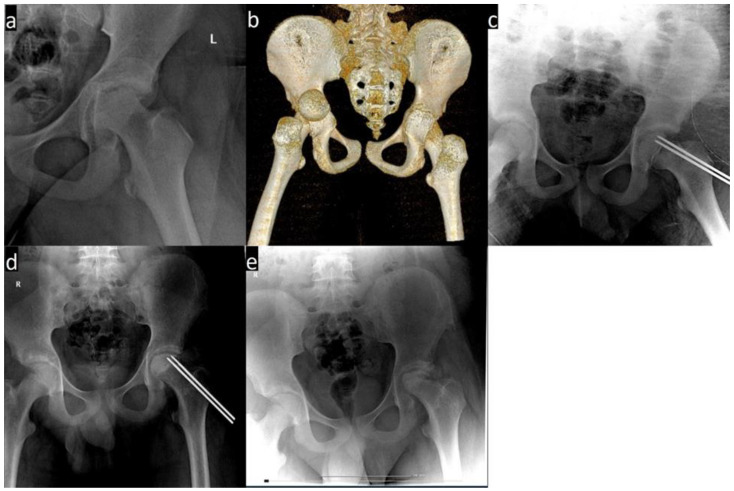
(**a**) X-ray showing a dislocation of the left hip associated with a transphyseal fracture. (**b**) Left hip fracture dislocation on a CT image. (**c**) Postoperative X-ray. (**d**) Signs of femoral head necrosis 5 months after surgery. (**e**) AP view of the hip 1 year after trauma.

**Figure 3 children-09-00612-f003:**
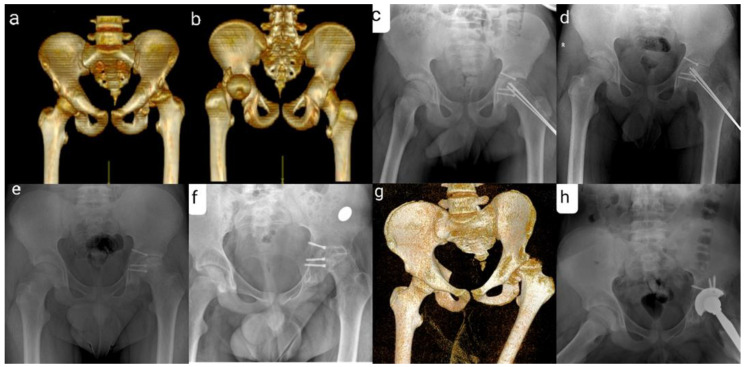
(**a**,**b**) Fracture-dislocation of the left hip associated with posterior column fracture of the acetabulum—CT image. (**c**) Postoperative X-ray. (**d**) Signs of femoral head necrosis 4 months after surgery. (**e**) X-ray of the pelvis after removal of the threaded Kirschner wires (**f**) Pelvic radiography 1 year and 8 months after surgery. (**g**) Pelvis CT 1 year and 8 months after surgery. (**h**) X-ray after total left hip arthroplasty.

**Table 1 children-09-00612-t001:** List of research articles.

Nr. Crt.	Article	Age (Years)	Nr	Gender	Trauma	Type of Dislocation	Interval from Injury to Reduction	Treatment	Complications	Outcome
**1.**	Herrera-Soto et al. [7]	13–15	5	1 F 4 M	road accident	5 P	-	5 ORIF	AVN	poor
**2.**	Odent el al. [8]	12–14	5	1 F 4 M	road accident	4 P 1 A	6 h	4 ORIF 1 ORIF	AVN	poor
**3.**	Palencia et al. [10]	12	1	M	road accident	P	5 h	ORIF	AVN	poor
**4.**	Kennon et al. [15]	11–15	12	2 F 10 M	11 sport accidents 1 road accident	9 P 3 A	-	11 ORIF 1 CR	AVN	poor
**5.**	Van Norwick et al. [17]	13	1	M	sport accident	P	9 h	ORIF	NA	good
**6.**	Forlin et al. [18]	11	1	F	road accident	-	-	ORIF	AVN *	poor
**7.**	Hougaard et al. [19]	13, 16	2	2 M	road accident	2 P	1 patient 4 days 1 patient 24 h	ORIF	AVN	poor
**8.**	Schoenecker et al. [20]	13, 15	2	2 M	1 altercation 1 sport accident	2 P	1 patient over 6 h	ORIF	1 AVN ** 1 NA	1 patient poor 1 patient good
**9.**	Novais et al. [21]	14	1	M	sport accident	P	7 days	ORIF	NA	good
**10.**	Basaran et al. [22]	10	1	M	road accident	P	16 h	ORIF	AVN ***	poor
**11.**	Nazareth et al. [23]	13	1	M	sport accident	-	-	ORIF	NA	good

M = male; F = female; P = posterior; A = anterior; ORIF = open reduction with internal fixation; CR = close reduction; NA = nonassociated; AVN = avascular necrosis; * associated premature physeal closure and chondrolysis; ** associated subluxation of the femoral head; *** associated arthritis and femoral head subluxation.

**Table 2 children-09-00612-t002:** List of patients from the 11 articles and the cases treated in our clinic.

Nr. Crt.	Age (Years)	Gender	Other Injury	Type of Treatment	Type of Approach	Complications	Time to AVN (Months)
1.	13	F	bilateral tibiae fractures	ORIF with 3 Kirschner wires	posterolateral	AVN	13
2.	14	M	NA	ORIF with 2 or 3 screws	posterolateral	AVN	15
3.	13	M	distal radius fracture	ORIF with 2 or 3 screws	posterolateral	AVN	9
4.	15	M	NA	ORIF with 2 or 3 screws	posterolateral	AVN	3
5.	14	M	NA	ORIF with 2 or 3 screws	greater trochanteric osteotomy	AVN	4
6.	12	F	NA	ORIF with 1 screw	posterior	AVN	6
7.	14	M	NA	ORIF with 2 screws	posterior	AVN	6
8.	13	M	NA	ORIF with 2 screws	posterior	AVN	6
9.	14	M	NA	ORIF with 2 screws	anterolateral	AVN	6
10.	14	M	NA	ORIF with 2 screws	posterior	AVN	6
11.	12	M	left anterior column fracture of the acetabulum	ORIF with 2 screws	posterolateral	AVN	6
12.	14	M	NA	ORIF	posterior	AVN	7–15
13.	12	M	NA	ORIF	posterior	AVN	7–15
14.	14	M	NA	ORIF	posterior	AVN	7–15
15.	12	M	NA	ORIF	posterior	AVN	72
16.	12	M	NA	ORIF	posterior	AVN	7–15
17.	11	F	multiple trauma severe head injury	CR	-	AVN	7–15
18.	14	M	NA	ORIF	posterior	AVN	7–15
19.	14	M	NA	ORIF	posterior	AVN	7–15
20.	15	M	NA	ORIF	posterior	AVN	48
21.	15	M	NA	ORIF	anterior	AVN	7–15
22.	12	F	NA	ORIF	anterior	AVN	7–15
23.	14	M	NA	ORIF	anterior	AVN	7–15
24.	13	M	anterior femoral head fracture	ORIF with 2 screws	greater trochanteric osteotomy	NA	-
25.	11	F	NA	ORIF	-	AVN *	-
26.	13	M	bilateral tibiae and fibular fractures, peroneal nerve paralysis, acetabulum rim fracture	ORIF with pins	posterior	AVN	24
27.	16	M	acetabulum rim fracture	ORIF with Smith-Peterson nail	posterior	AVN	18
28.	13	M	NA	ORIF with 2 Kirschner wires	greater trochanteric osteotomy	NA	-
29.	15	M	NA	ORIF with 2 screws	posterolateral	AVN **	5
30.	14	M	NA	ORIF with 3 screws	posterolateral	NA	-
31.	10	M	NA	ORIF with 3 retrograde Herbert screws	anterior	AVN ***	3
32.	13	M	NA	ORIF with screws	greater trochanteric osteotomy	NA	-
33.	14	M	NA	ORIF with 2 Kirschner wires	anterolateral	AVN	2
34.	12	M	NA	ORIF with 2 Kirschner wires	posterolateral	AVN	5
35.	14	M	anterior column fracture of the acetabulum	ORIF with 3 Kirschner wires	posterolateral	AVN	4

M = male; F = female; ORIF = open reduction with internal fixation; CR = close reduction; NA = nonassociated; AVN = avascular necrosis; * associated premature physeal closure and chondrolysis; ** associated subluxation of the femoral head; *** associated arthritis and femoral head subluxation.

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
