# Peer review of "Traumatic Hip Dislocation Associated with Proximal Femoral Physeal Fractures in Children: A Systematic Review"

_children, 2022, doi:10.3390/children9050612_

Round 1
Reviewer 1 Report
The study entitled “Traumatic Hip Dislocation Associated with Proximal Femoral Physeal Fractures in Children. A Systematic Review” is a systematic review and case series to describe the complications caused by hip dislocations associated with transphyseal femoral neck fractures.
My suggestions are as follows:
1. Title should be updated to reflect the study more appropriately and should include the "case series" in the time as well.
"Traumatic Hip Dislocation Associated with Proximal Femoral Physeal Fractures in Children. A Systematic Review and Case Series"
2. Introduction section of eth article should put the context to this current research and justify the need for performing this study. Generally, the last paragraph of the introduction section says something like this, " First provide the main driving reasons to perform this study, and then say. Hence, we performed this systematic review and case series......"
3. List the database searched in Box 1 of Figure 1.
4. "We used Covidence to review", this is mentioned twice. Please remove one mention.
5. Discussion Section should start with summarising the key finding from this study and then should proceed with discussing the relevant observation in teh context of existing literature.
6. the percentage of femoral head osteonecrosis after traumatic hip dislocation associated with transphyseal femoral neck fracture is 88,57%.
This 88,57% number is nowhere mentioned in the result section. It seems to be important as per the author as they mentioned this in the abstract as well. Hence it will be great if this is captured in the result section as well.
7. "not having many articles presenting this type of injury" this is not a limitation of this study, hence should be removed from the limitation section.
8. Similarly, "included studies not having complete information" is also not a limitation of this study.
9. The actual limitation of this study is the lack of registered protocol. This systematic review has not been registered online. Please add this point to the limitation.
10. Please also include that no grey literature (conference abstracts) were searched in the limitation section.
11. Who are the two independent researchers? Put their initials in the method section/description text.
12. The assessment of the risk of bias is an integral part of any high-quality systematic review. The authors should assess and report the risk of bias in the included studies.
The author should also complete and provide the PRISMA checklist to facilitate the assessment around transparent reporting of the present systematic review.
12. "Traumatic hip dislocation associating transphyseal femoral neck fracture in children is a rare injury with very poor prognosis, with an incidence of almost 100% for AVN."
This is the finding from this systematic review. The conclusion should highlight the finding from this systematic review rather than finding from any one single paper that is included in the review.
Author Response
Thank you for your very insightful suggestion on our paper. We have made some changes as explain bellow accordingly with the received comments. We have highlighted in yellow the modified sentences. We have also made an extended English check and we have rewritten some paragraphs.
The study entitled “Traumatic Hip Dislocation Associated with Proximal Femoral Physeal Fractures in Children. A Systematic Review” is a systematic review and case series to describe the complications caused by hip dislocations associated with transphyseal femoral neck fractures.
My suggestions are as follows:
- Title should be updated to reflect the study more appropriately and should include the "case series" in the time as well.
"Traumatic Hip Dislocation Associated with Proximal Femoral Physeal Fractures in Children. A Systematic Review and Case Series"
Author response: Thank you for pointing this out, we have added this information in the abstract also.
- Introduction section of eth article should put the context to this current research and justify the need for performing this study. Generally, the last paragraph of the introduction section says something like this, " First provide the main driving reasons to perform this study, and then say. Hence, we performed this systematic review and case series......"
Author response: Thank you for this suggestion, we have add at the end of the Introduction section a paragraph describing the need for this study.
- List the database searched in Box 1 of Figure 1.
Author response: We listed the data based in the first paragraph of the section Materials and Methods. We conducted the research on PubMed, Google Scholar and Web of Science and uploaded the results in Covidence.
- "We used Covidence to review", this is mentioned twice. Please remove one mention.
Author response: Thank you for pointing out this mistake, we have removed the sentence from one of the paragraphs.
- Discussion Section should start with summarizing the key finding from this study and then should proceed with discussing the relevant observation in the context of existing literature.
Author response: We have added a paragraph summarizing the content of our result to the Discussion Section.
- The percentage of femoral head osteonecrosis after traumatic hip dislocation associated with transphyseal femoral neck fracture is 88,57%.
This 88,57% number is nowhere mentioned in the result section. It seems to be important as per the author as they mentioned this in the abstract as well. Hence it will be great if this is captured in the result section as well.
Authors response: Thank you for pointing out this mistake, we have corrected that and added this information in the results section.
- "not having many articles presenting this type of injury" this is not a limitation of this study, hence should be removed from the limitation section.
- Similarly, "included studies not having complete information" is also not a limitation of this study.
- The actual limitation of this study is the lack of registered protocol. This systematic review has not been registered online. Please add this point to the limitation.
- Please also include that no grey literature (conference abstracts) were searched in the limitation section.
Author response: Thank you very much for the notes regarding the limitation of our review and We thing that it has helped us better state the nature of our study. We have removed the to limitation and added the suggested one.
- Who are the two independent researchers? Put their initials in the method section/description text.
Author response: We have added the initials in the method section.
- The assessment of the risk of bias is an integral part of any high-quality systematic review. The authors should assess and report the risk of bias in the included studies.
The author should also complete and provide the PRISMA checklist to facilitate the assessment around transparent reporting of the present systematic review.
Author response: Two reviewers independently assess the risk of bias using the standard Cochrane Risk of Bias domains, none of the studies was judged to have a significant risk of bias.
We have fallowed the PRISMA checklist that apply to our study.
- "Traumatic hip dislocation associating transphyseal femoral neck fracture in children is a rare injury with very poor prognosis, with an incidence of almost 100% for AVN."
This is the finding from this systematic review. The conclusion should highlight the finding from this systematic review rather than finding from any one single paper that is included in the review.
Author response: This is right we have rewritten the conclusion to be related with our one result.
Thank you very much for all your suggestions and we hope our changes will meet your requirements.
Reviewer 2 Report
This is a nice and interesting manuscript concerning hip dislocation with concomitant proximal physeal fracture of the proximal femur resp. of the pelvis. Only adolescents are introduced in this review with a combination of three patients treated by the authors. The mechanism of injury was only a high energy trauma. In nearly 90% resulted a avascular necrosis of the femoral head. This is a well-known complication in transphyseal femoral neck fracture. In adolescents principles of adult orthopedic surgery must be applied. After the age of 12 years there will be no relevant growth at the proximal growth plate of the femur, so stable ORIF has to be recommended. The authors however did apply 2 or 3 K-wires, which in the first case seem to be placed partially outside the femoral neck and without a proper reduction, still leaving some gap. This case did develop AVN in an early stage. The authors should comment on the fact why they did not use screws for more stable anchorage. Furthermore the risk for AVN could be increased by unstable ORIF. This needs a comment as well.
Author Response
Thank you very much for your kind comments regarding our paper, we agree that AVN is a well -known complication regarding this injury but after the poor prognostic of the series of cases treated in our clinic, we wanted to better understand this type of injury and to find the best treatment with the best success rate and that is why we decided to study this injury. We have added that in the introduction to be clearer about this.
Regarding the use of screw compared with pins, in our systematic review we do not have all the data to assess this factor. We have data regarding screws or pins for 19 cases and 16 were treated with screws from which 13 developed AVN.
In the first presented case the pins are inside the femoral neck and head, the pictures may not very clear. We used partial threaded pins to secure the fixation. We did not have any hardware displacement in this case or second displacement of the femoral head.
The use of pins could be a reason for AVN, that requires further research. In case of SCFE from the literature there is no difference regarding AVN between cases treated with pins and cases treated with screws, but pins have more complication and revision rate. We have added a paragraph with this idea in the discussion section.
Round 2
Reviewer 1 Report
Thank you for incorporating the suggestions.
This manuscript is a resubmission of an earlier submission. The following is a list of the peer review reports and author responses from that submission.
Round 1
Reviewer 1 Report
There is not an omogeneus treatment of the cases, tha surgical approach is not clearly definited: in the first case which one was the surgical approach? in the second a lateral approach, in the third a Kocher Langenbecck,
It's not so clear the message that you want to give us: traumatic hip dislocarion in children is a very rare condition in children and even if you treat it as soon as possible the risk of complication is very high'? everyone knows this.
In the first case there were not a fracture before the reductrion: it's a iatrogenic fracture.
I think that you have to explain the surgical treatment of prof. Ganz for the surgical luxation of the hip with the Dunn modified approach: the preservation of the posterior vascular pedunculum is the only way to avoid the necrosis of the femoral head.
Author Response
There is not an omogeneus treatment of the cases, tha surgical approach is not clearly definited: in the first case which one was the surgical approach? in the second a lateral approach, in the third a Kocher Langenbecck,
Response- Thank you for taking your time to revise our paper.
Thank you for your comment and the opportunity to better explain the surgical approach. The three patients were treated by different surgeons and the choice of the approached was made by considering the position of the hip luxation if it is associated with fracture of the acetabulum and the surgeon experience. We do not think that there is a gold standard approach. Most of the cases in the literature are treated by these types of approaches. We have rewrite some of the treatments to better understand our ideas.
It's not so clear the message that you want to give us: traumatic hip dislocation in children is a very rare condition in children and even if you treat it as soon as possible the risk of complication is very high'? everyone knows this.
Response We tried to emphasis the idea that this type of injury has complication like AVN in almost all the patients even though the approach is different, and the surgery is done by different surgeons. All the cases that had hip luxation and physeal separation developed AVN regarding the treatment.
In the first case there were not a fracture before the reduction: it's a iatrogenic fracture
Response- Thank you for pointing this out, we have rewritten the presentation of this case to be clearer. The reduction maneuvers were very gentle and were more small manipulation and the phiseal separation happened at the firs maneuver. We consider that there was a lesion of the growth cartilage after the luxation that led to the phiseal separation. We considered that this is a very interesting case that better emphasis the necessity of fixation before reduction in these cases.
I think that you have to explain the surgical treatment of prof. Ganz for the surgical luxation of the hip with the Dunn modified approach: the preservation of the posterior vascular pedunculum is the only way to avoid the necrosis of the femoral head.
Response -In our clinic this procedure was done only for SCFE, we did find one case in the literature that was treated with Dunn modified procedure and we have written this in the text. This is a very interesting idea and for cases with no acetabular fracture this could be a better solution.
Reviewer 2 Report
The authors did a study, named: “Traumatic Hip Dislocation Associated with Proximal Femoral Physeal Fractures in Children”.
It is stated that this is a retrospective study, but in methods it is clearly stated that PubMed was searched for articles. It is unclear to me the type of this study.
My concerns are as follow:
- Abstract:
- transepiphyseal femoral neck fracture? Did the authors want to write: transphyseal?
- please change:” patients of the age over 11 years old” into: “patients over 11 years of age”.
- preseted should be presented
- AVN acronym is stated without prior setting in brackets as an explanation for avascular necrosis
- Introduction:
- THD abbreviation is not previously properly explained!
- should be more covered and widened! Maybe to write something about postreduction or postoperative period and treatment. There is an option of punction of acetabular joint and extraction and elimination of hemathoma, etc... As this is a rare condition all treatment options should be listed and referenced!
- Methods:
- table or something else with demographic data, and other input data should be placed in methods
- Please clearly state inclusion and exclusion criteria, as well as primary and secondary study outcomes.
- This is the part i do not understand: “We carried research on PubMed using the following terms: traumatic hip dislocation 83 in children, fracture-dislocation of the hip in children and dislocation of the hip associated 84 with transepiphyseal fracture in children”.
If the authors performed a review, it is not satisfactory to go through PubMed alone, because it represent mostly USA region. Please include in the search WoS or Scopus even.
Other thing, why was the exclusion age under 11 years in this search? Furthermore, maybe a search based on adjective transepiphyseal instead of transphyseal, or physeal gave false negative results. I would recommend consulting a librarian about this.
- I would very much like the Authors to put something of the follow up protocol (X-rays, exams, MRI) as well as postoperative period and management.
- My opinion is that the operation, closed reduction and internal fixation, should be better put in methods then in results.
- Results:
- mean age, and other “mean” data should be presented as medians.
- should better be raw data with p values, maybe a table or a figure?
- Results are too much descriptive as said before! There is a mix of data for methods and for discussion.
- The table in Results (review table) is named table 2, but I can’t find table 1!
- Operative figures should be put in methodology.
- Discussion:
- Please avoid repetition from Introduction in discussion. Discussion should be strictly stuctured!
This is some mix betwen retrospective study, based on 3 patients, that more looks like a case reports, and a partial review. The study is interesting and but should be modified as a specific type of study and to make more reader friendly.
Author Response
The authors did a study, named: “Traumatic Hip Dislocation Associated with Proximal Femoral Physeal Fractures in Children”.
It is stated that this is a retrospective study, but in methods it is clearly stated that PubMed was searched for articles. It is unclear to me the type of this study.
Response Thank you for revising our paper and for all the comments that helped improve our manuscript.
We attempted to do a review of the literature regarding this subject, we retrospectively study the three cases in our clinic. We have rewritten this at the beginning of the Methods .
My concerns are as follow:
- Abstract:
- transepiphyseal femoral neck fracture? Did the authors want to write: transphyseal?
- please change:” patients of the age over 11 years old” into: “patients over 11 years of age”.
- preseted should be presented
- AVN acronym is stated without prior setting in brackets as an explanation for avascular necrosis
Response- Thank you for pointing this out. We agree with this comment. Therefore, we have modified in the whole article the word transepiphyseal with the word transphyseal. We use the word transepiphyseal according to Delbet and Colonna classification use in article 9 (Shaath MK, Shah H, Adams MR, Sirkin MS, Reilly MC. Management and Outcome of Transepiphyseal Femoral Neck Fracture-Dislocation with a Transverse Posterior Wall Acetabular Fracture: A Case Report. JBJS Case Connect. 2018)
We have, accordingly, changed “patients of the age over 11 years old” into “patients over 11 years of age”, we changed preseted with presented and we explained the AVN acronym and replaced avascular necrosis with AVN in the whole article.
- Introduction:
- THD abbreviation is not previously properly explained!
- should be more covered and widened! Maybe to write something about postreduction or postoperative period and treatment. There is an option of punction of acetabular joint and extraction and elimination of hemathoma, etc... As this is a rare condition all treatment options should be listed and referenced!
Response - We explained the THD abbreviation in the first use and used in the entire text
Regarding the treatment options we have rewritten parts of the introduction and presented more treatment options and follow up depending on the age of the patient. We did not discussed the puncture of the joint to remove hematoma because we tried to focused on hup traumatic luxation with separation of the femoral head that requires open reduction.
- Methods:
- table or something else with demographic data, and other input data should be placed in methods
- Please clearly state inclusion and exclusion criteria, as well as primary and secondary study outcomes.
Respone- Thank you for pointing this out the absence of table 1. We added the table with the demographic data of all the patients.
We offered explication about primary and secondary study outcome and also we clarified the inclusion and exclusion criteria.
The inclusion criteria were age over 11 years of age, hip dislocation with phiseal separation, articles after 2000.
The exclusion criteria were patient under 11 years old, patient with only THD or patient with only transphyseal femoral head and neck fracture, patient with other systemic condition and articles before year 2000. We also excluded all the articles in languages other than English.
The primary outcome in our study was being able to determine the rates of developing osteonecrosis after this type of injury. The secondary outcome in our study was finding the correlation between osteonecrosis and patient age, sex and associated acetabular fracture.
- This is the part i do not understand: “We carried research on PubMed using the following terms: traumatic hip dislocation 83 in children, fracture-dislocation of the hip in children and dislocation of the hip associated 84 with transepiphyseal fracture in children”.
If the authors performed a review, it is not satisfactory to go through PubMed alone, because it represent mostly USA region. Please include in the search WoS or Scopus even.
Other thing, why was the exclusion age under 11 years in this search? Furthermore, maybe a search based on adjective transepiphyseal instead of transphyseal, or physeal gave false negative results. I would recommend consulting a librarian about this.
Respone-We had researched WOS also, but we did not find any new article.
This aspect regarding the search could be true but we also used traumatic hip dislocation in children and verified all of this articles. We searched with both transepiphyseal and transphyseal but we used transepiphyseal in the text because of the article mentioned above.
We used the exclusion criteria “age under 11 years old” because of Truetta article(TRUETA J. The normal vascular anatomy of the human femoral head during growth. J Bone Joint Surg Br. 1957). In this article he stated that there are five phases of femoral head vascularity from birth to maturity. The fourth stage (nine to ten years) is characterized by the fact that the physis continues to act like a barrier, but there are vessels that emerges from ligamentum teres, penetrates the epiphysis and makes anastomosis. We studied the patients that according to Truetta were included in the fifth stage in which the adult stage of vascularization of the femoral head begins to form, this vascularization is represented by anastomosis between lateral epiphyseal vessels, metaphyseal vessels, and ligamentum teres vessels. This anastomosis penetrates the growing plate. It is possible that this is reason for which transphyseal femoral head fracture at adolescent lead to osteonecrosis, due to the shearing of the vessels and local ischemia.
It could be interesting to compare age groups for AVN.
- I would very much like the Authors to put something of the follow up protocol (X-rays, exams, MRI) as well as postoperative period and management.
As recommended, we included in this section the follow up protocol as well as postoperative period and management in the methods.
- My opinion is that the operation, closed reduction and internal fixation, should be better put in methods then in results.
Thank for pointing out the close reduction we have described this treatment and explained the role of the close reduction and pinning prior to this.
- Results:
- mean age, and other “mean” data should be presented as medians.
- should better be raw data with p values, maybe a table or a figure?
- Results are too much descriptive as said before! There is a mix of data for methods and for discussion.
- The table in Results (review table) is named table 2, but I can’t find table 1!
- Operative figures should be put in methodology.
Respone - We have redone some of the calculation and presented the age as a median also. We presented the detailed data in the text. We moved the figures to methods.
We have resolved with table 1 and we introduced in the text, is a demographic data table. We moved the figures in Methods.
- Discussion:
- Please avoid repetition from Introduction in discussion. Discussion should be strictly stuctured!
Response- We have rewritten a part of the discussion and removed the repetitive part. We have tried to better point out our ideas in the discussions and in the conclusions.
This is some mix betwen retrospective study, based on 3 patients, that more looks like a case reports, and a partial review. The study is interesting and but should be modified as a specific type of study and to make more reader friendly.
Response Our intension is to write a review of the literature regarding this subject and retrospectively study our cases and include them in all the data because is a rare condition and there are not so many cases reported.

Round 2
Reviewer 1 Report
I think you can inprove your English.
YOU HAVE TO EXPLAIN BETTER THE PURPOSE of rhis study, which desceibes some cases treated in different
ways
Author Response
We have rewritten some part to better describe the purpose of our study.
We have improved our English and corrected some of the expressions.
The purpose of our study was to do a review of the literature for traumatic hip dislocation in combination with transphyseal separation of the femoral head and to add to this the cases treated in our clinic.
After our poor results we tried to look for corelation between the treatment, demographic data, and AVN. Unfortunately, all the cases that we found in literature developed AVN.
Thank you for reviewing our paper and for giving us a chance to better explain the purpose of this.
Reviewer 2 Report
With all respect to authors and to corrections made, I stand as before that the setting of the paper is not right. This kind of paper is not reader friendly.
I would recommend doing the case reports...
Author Response
Thank you for reviewing our paper.
We have rewritten some parts of the paper and corrected the English to the paper more reader friendly.
For this paper we conducted a literature review and added the three cases treated in our clinic. The purpose of this study was to try and find in the literature if different treatment or a different approach will have a different result. Unfortunately, all the literature cases developed AVN.
We decided to add our cases also to our review because of the small number of cases that we find in the literature.